# From Acute to Chronic: Unraveling the Pathophysiological Mechanisms of the Progression from Acute Kidney Injury to Acute Kidney Disease to Chronic Kidney Disease

**DOI:** 10.3390/ijms25031755

**Published:** 2024-02-01

**Authors:** Tzu-Hsuan Yeh, Kuan-Chieh Tu, Hsien-Yi Wang, Jui-Yi Chen

**Affiliations:** 1Division of Nephrology, Department of Internal Medicine, Chi Mei Medical Center, Tainan 71004, Taiwan; leaf9970040@gmail.com (T.-H.Y.); hsienyiwang@gmail.com (H.-Y.W.); 2Division of Cardiology, Department of Internal Medicine, Chi Mei Medical Center, Tainan 71004, Taiwan; rbobby2050@hotmail.com.tw; 3Department of Sport Management, College of Leisure and Recreation Management, Chia Nan University of Pharmacy and Science, Tainan 71710, Taiwan; 4Department of Health and Nutrition, Chia Nan University of Pharmacy and Science, Tainan 71710, Taiwan

**Keywords:** acute kidney injury, acute kidney disease, biomarker, chronic kidney disease

## Abstract

This article provides a thorough overview of the biomarkers, pathophysiology, and molecular pathways involved in the transition from acute kidney injury (AKI) and acute kidney disease (AKD) to chronic kidney disease (CKD). It categorizes the biomarkers of AKI into stress, damage, and functional markers, highlighting their importance in early detection, prognosis, and clinical applications. This review also highlights the links between renal injury and the pathophysiological mechanisms underlying AKI and AKD, including renal hypoperfusion, sepsis, nephrotoxicity, and immune responses. In addition, various molecules play pivotal roles in inflammation and hypoxia, triggering maladaptive repair, mitochondrial dysfunction, immune system reactions, and the cellular senescence of renal cells. Key signaling pathways, such as Wnt/β-catenin, TGF-β/SMAD, and Hippo/YAP/TAZ, promote fibrosis and impact renal function. The renin–angiotensin–aldosterone system (RAAS) triggers a cascade leading to renal fibrosis, with aldosterone exacerbating the oxidative stress and cellular changes that promote fibrosis. The clinical evidence suggests that RAS inhibitors may protect against CKD progression, especially post-AKI, though more extensive trials are needed to confirm their full impact.

## 1. Introduction

Impaired renal function is a disease with a broad spectrum in terms of the duration and progression of renal deterioration. Broadly, this can be divided into three stages: acute kidney injury (AKI), acute kidney disease (AKD), and chronic kidney disease (CKD) [1]. According to the 2012 KDIGO guidelines, the diagnosis of AKI could be defined as a 1.5-fold increased serum creatinine level from baseline or less urine output (<0.5 ml/kg/h) within 7 days [2]. AKD is defined as acute or subacute damage and/or loss of kidney function for between 7 and 90 days [3]. CKD is defined as abnormalities of kidney structure or function, present for more than 3 months, with implications for health [4]. Nowadays, kidney disease can be classified as the continuum of AKI, AKD, and CKD [5].

Those with AKI have a higher risk of AKD, CKD, or even progression to end-stage kidney disease (ESKD) [6]. It is believed that there is a causal relationship between AKI and CKD [7,8]. A cohort study revealed that 24.6% of patients with AKI developed CKD within a 3-year follow-up period [9]. A meta-analysis found that the pooled rate of CKD development among patients with AKI is 25.8 per 100 person-years [8]. Moreover, it is also believed that patients with AKD have a higher risk for progression to CKD [10]. One meta-analysis, encompassing a large cohort of 1,114,012 patients with AKD, demonstrated that 37.2% of these patients are likely to progress to CKD. This finding underscores that the risk of developing CKD is notably higher in patients with AKD compared to those without AKD [11].

With its rising incidence, the economic and healthcare burden caused by AKI and its complications is increasing [12]. To prevent the progression of renal dysfunction and further complications from AKI, the early detection of AKI has become an important issue in clinical practice. Nevertheless, the routine diagnostic test for AKI is serum creatinine, which is a delayed and unreliable biomarker for AKI due to various reasons [13]. It remains a clinical dilemma to directly evaluate the etiologies of AKI without resorting to renal biopsy despite numerous biomarkers having been proposed within recent decades as indicators for the specific site of nephron damage, including neutrophil gelatinase-associated lipocalin (NGAL), cystatin C, liver-type fatty acid-binding protein (L-FABP), kidney injury molecule-1 (KIM-1), and interleukin 18 (IL-18), among others [14,15]. Moreover, the progression of AKI to CKD involves numerous complicated molecular mechanisms (basically the consequence of cellular injury and maladaptive repair), which are still not well elucidated [16,17]. Hence, the current review focuses on the role of biomarkers including NGAL, tissue inhibitor of metalloproteinases 2 X insulin-like growth factor binding protein 7 ([TIMP-2]X[IGFBP7]), KIM-1, and L-FABP in the early detection of AKI, the pathophysiology of AKI, and the molecular pathways involved in the AKI to CKD transition.

## 2. Biomarkers of AKI

Extensive research is currently underway to explore novel biomarkers for the early detection and prognosis of AKI. As outlined by the Acute Disease Quality Initiative Consensus Conference on AKI biomarkers, these biomarkers are categorized into three main categories: stress markers, damage markers, and functional markers [18]. Stress markers serve as early indicators of cellular stress, allowing for the prediction of AKI. Conversely, damage markers indicate structural damage, which may or may not result in a reduction of renal function. Lastly, functional markers are linked to changes in glomerular filtration, thereby offering a measure of renal function alterations. Subclinical AKI is defined as the presence of at least one positive novel biomarker without elevated serum creatinine levels, indicating early kidney stress and damage before functional loss. The added value of these new biomarkers lies in their ability to enable an earlier diagnosis and to detect kidney injury, even without evident dysfunction [19,20,21].

### 2.1. Stress Markers

Urinary Dickkopf-3 (DKK3), a glycoprotein originating from kidney tubular epithelial cells (TECs), is utilized in the risk assessment and prediction of AKI. Preoperative levels of urinary DKK3 have been identified as an independent predictor for the occurrence of postoperative AKI [22]. The urinary levels of TIMP-2 and IGFBP-7 serve as markers indicating G1 cell cycle arrest. These markers may show a rapid increase after cellular stress, typically within 4 to 12 h, even before the occurrence of injury [23,24]. In patients progressing to AKI stages 2–3, the concentrations of urinary [TIMP-2] • [IGFBP7] surged on the day of exposure, displaying a distinct pattern of rise and subsequent decline surrounding the majority of exposures [25]. Consequently, biomarkers such as TIMP-2 and IGFBP-7 are instrumental in detecting subclinical forms of AKI that cannot be identified using conventional testing methods.

### 2.2. Damage Markers

Alanine aminopeptidase, alkaline phosphatase, and γ-glutamyl transpeptidase are enzymes located on the brush border villi of proximal tubular cells [24]. When these cells sustain damage, these enzymes are released into the urine and can be detected and used to indicate tubular injury in patients [26].

The renin–angiotensin–aldosterone system (RAAS) is activated in AKI, leading to increased angiotensin II (Ang II) levels in the kidney. This Ang II triggers pro-inflammatory and pro-fibrotic pathways, contributing to the progression of AKI [27]. Angiotensinogen, the precursor polypeptide for angiotensin peptides, is stable in urine, making it a practical marker for assessing RAAS activity, as compared to directly measuring Ang II in urine, and it offers more specific insights. Urinary angiotensinogen is suggested as a marker for intrarenal RAAS activity and forecasts the advancement of CKD. In patients with critical AKI, angiotensinogen effectively predicted AKI worsening (area under the ROC curve (AUC) = 0.77) and the combined outcome of requiring renal replacement therapy (RRT) or mortality (AUC = 0.73) [28].

Meprin A, consisting of α- and β-subunits, is a membrane-associated neutral metalloendopeptidase in the astacin family of zinc endopeptidases, predominantly located in the brush-border membranes of proximal tubules and intestines [29]. Extensive research has highlighted its role in the pathogenesis of AKI caused by ischemia–reperfusion (IR) injury and cisplatin nephrotoxicity. Notably, after renal IR, a significant shift in meprin A’s distribution from its regular linear pattern along brush-border membranes to the underlying basement membrane has been observed. This redistribution of meprin A during IR is believed to contribute to cellular damage and incites an inflammatory response. Moreover, the presence of meprin A in urine during AKI indicates its shedding under such pathological conditions [30,31,32].

Calprotectin is a cytosolic calcium-binding complex derived from neutrophils and monocytes. In cases of intrinsic AKI, the levels of calprotectin in urine are significantly elevated [33,34]. C–C motif chemokine ligand 14 (CCL14) is a pro-inflammatory chemokine released into urine in response to stress or damage to tubular cells. According to findings from the RUBY study, elevated levels of CCL14 serve as a predictive marker for persistent AKI in critically ill patients, particularly those with severe AKI [35]. NGAL exists in three different types: a monomeric glycoprotein form derived from neutrophils and TECs, a homodimeric protein originating from neutrophils, and a heterodimeric protein produced by tubular cells. These forms of NGAL can be detected in both serum and urine during the development of AKI, particularly after ischemic or toxicity-induced damage to the kidney [26,33,36], and NGAL had the best predictive accuracy for the occurrence of AKI [37].

Urine KIM-1, a transmembrane glycoprotein produced by proximal tubular cells, is released into the urine following tubular damage. It has been established as a proven marker of AKI in adults [15,36]. Indeed, additional biomarkers are released in response to tubular damage, including L-FABP, a protein located in the cytoplasm of renal proximal tubules. IL-18, a pro-inflammatory cytokine, is also among the biomarkers associated with tubular damage. Monitoring the levels of these biomarkers can contribute to the assessment and diagnosis of AKI [36,38]. These molecules consist of constitutive proteins released by the damaged kidney, substances upregulated in response to injury, and products from non-kidney tissues that are subjected to filtration, reabsorption, or secretion by the kidney [18].

### 2.3. Functional Markers

The levels of cystatin C, a cysteine protease inhibitor produced by nucleated human cells, are increased within 12–24 h following renal injury [39]. Cystatin C is considered to have a better accuracy than serum creatinine in identifying individuals with a reduced glomerular filtration rate (GFR) [40]. Firstly, serum creatinine is not capable of promptly reflecting changes in the glomerular filtration rate, especially in situations where the GFR is not in a steady state [41]. Additionally, the clearance of serum creatinine from the body is not solely through glomerular filtration; it also involves partial secretion by the renal tubules. This widely recognized process can lead to a significant overestimation of the GFR. Therefore, cystatin C is particularly useful for detecting even mild declines in GFR [42].

Proenkephalin A is an endogenous polypeptide hormone found in various tissues, such as the adrenal medulla, nervous system, immune system, and renal tissue [43]. It has been reported that proenkephalin A is a useful biomarker for the early detection of AKI and predicting a shorter duration and successful liberation from RRT [44,45]. Both cystatin C and proenkephalin A are considered damage biomarkers, as well as functional biomarkers. When used in combination, they contribute to a more comprehensive assessment and accurate diagnosis of AKI [24] (Table 1).

### 2.4. Limitations and Possibilities for Future Studies of Novel AKI Biomarkers

Unlike myocardial troponin, which is specifically released by damaged myocardial muscle cells [46], kidney injury biomarkers and serum creatinine may potentially interact with the inflammatory response. This cross-reaction can complicate their ability to accurately reflect kidney damage, particularly under conditions such as sepsis [47].

The peak concentrations of different biomarkers vary based on the timing after the initial insult, which presents a significant challenge in interpreting these findings accurately within a clinical setting. Future research should emphasize exploring the mechanisms of injury to enhance our comprehension of various AKI phenotypes that is grounded in their pathophysiological characteristics. Exploring the biological transition from initial kidney stress to the emergence of subclinical or clinical AKI could open up novel targets for therapeutic interventions. These advancements may encompass focused treatments or strategic approaches, all with the ultimate goal of improving patient outcomes.

**Table 1 ijms-25-01755-t001:** Biomarkers for acute kidney injury.

Types of Marker	Marker	Clinical Application
Stress marker	Urine	
	DKK3	Preoperative levels of urinary DKK3 have been identified as an independent predictor for the occurrence of postoperative AKI [22]. AUC for postoperative AKI: 0.783 [22].
	TIMP-2 IGFBP-7	These markers may show a rapid increase after cellular stress, typically within 4 to 12 h, even before the occurrence of injury [23,24]. AUC for AKI prediction: 0.80 in ICU patients [23]; 0.84 in patients who underwent cardiac surgery [48].
Damage marker	Urine	
	Alanine aminopeptidase	Diagnostic relevance in nephrolithiasis [49]. Positive correlation between urinary alanine aminopeptidase concentrations and glomerulonephritis [50].
	Alkaline phosphatase	Endre et al. used alkaline phosphatase as a biomarker of acute kidney injury in the EARLYARF trial [51]. AUC for diagnosis of AKI: 0.45 in ICU patients [52].
	γ-glutamyl transpeptidase	The Translational Research Investigating Biomarker Endpoints in AKI (TRIBE-AKI) study evaluated the role of γ-glutamyl transpeptidase in AKI diagnosis [53].
	Calprotectin	Calprotectin is an indicator for primary intrinsic AKI [54]. AUC for diagnosis of intrinsic acute renal failure: 0.97 [55].
	CCL14	Predictive marker for persistent AKI in critically ill patients in the RUBY study [35]. AUC for predicting persistent severe AKI: 0.81 [56].
	NGAL	Elevated levels of urinary NGAL are useful for predicting AKI, differentiating intrinsic AKI from pre-renal AKI, and predicting renal non-recovery, in-hospital mortality, and long-term CKD progression [57]. AUC for AKI prediction: 0.87 in all hospitalized patients [58].
	Angiotensinogen	Predictive of AKI progression, particularly in the setting of decompensated heart failure [27]. AUC for AKI prediction: 0.83 in patients with acute or chronic renal injury [59].
	Meprin A	Meprin A is induced by IR and cisplatin nephrotoxicity [29].
	KIM-1	Indicator of renal tubular damage [60]. Elevated levels of KIM-1 in patients with AKI may manifest prior to histological changes [61]. AUC for AKI prediction: 0.85 in patients who underwent cardiac surgery [62].
	L-FABP	Indicator of ischemic or toxic insults that result in tubulointerstitial damage [63]. AUC for AKI prediction: 0.81 in patients who underwent cardiac surgery [64].
	IL-18	Indicator of severity of albuminuria and deterioration of kidney function and associated with diabetic nephropathy [65]. AUC for AKI prediction: 0.75 in ICU patients [66].
	Serum	
	NGAL	NGAL can be detected after ischemic or toxicity-induced damage to the kidney [26,33,36] and has the best predictive accuracy for the occurrence of AKI [37].
Functional marker	Serum	
	Cystatin C	Better accuracy than serum creatinine in identifying individuals with reduced GFR [40] and level was increased within 12–24 h following renal injury [39]. AUC for prediction of sustained AKI: 0.80 [67].
	Proenkephalin A	Proenkephalin A serves as a useful biomarker for early detection of AKI and predicting a shorter duration and successful liberation from renal replacement therapy [44,45]. AUC for detecting AKI: 0.82 [68].

Abbreviations: AUC, area under the ROC curve; CCL14, C–C motif chemokine ligand 14; DKK-3, Dickkopf-3; IGFBP-7, insulin-like growth factor binding protein 7; IL-18, interleukin 18; KIM-1, kidney injury molecule-1; L-FABP, liver-type fatty acid-binding protein; NGAL, neutrophil gelatinase-associated lipocalin; TIMP2, tissue inhibitor of metalloproteinase-2.

## 3. Pathophysiology of Acute Kidney Injury

AKI arises from various insults, such as renal hypoperfusion, sepsis, major surgery, immunological diseases affecting the kidney parenchyma, administration of radiocontrast or nephrotoxic agents, and post-renal causes [69]. The pathophysiology of AKI varies depending on numerous conditions, and the etiology of AKI can be classified as pre-renal, intrinsic, or post-renal [70]. Here, we provide a concise overview of this pathophysiology.

### 3.1. Pre-Renal Causes

In cases of renal hypoperfusion induced by hypovolemia, autoregulation and neurohumoral mechanisms are triggered to maintain the GFR. Nevertheless, persistent renal hypoperfusion can lead to sustained inadequate oxygen delivery and depletion of adenosine triphosphate (ATP), causing cellular injury to the epithelium [71]. This can subsequently activate inflammatory responses, induce endothelial injury, and ultimately result in renal damage [72,73].In sepsis, inflammatory cytokines can induce leukocyte activation, recruit neutrophils, and trigger endothelial injury and coagulation. Additionally, these inflammatory mediators may bind to specific receptors expressed by renal endothelial and tubule epithelial cells, causing direct injury [74]. The release of damage-associated molecular patterns (DAMPs) by damaged cells further contributes to vasodilation, increased vascular permeability, and a pro-thrombotic environment [75]. Furthermore, filtered DAMPs and pathogen-associated molecular patterns (PAMPs) may activate Toll-like receptor 2 (TLR2) and Toll-like receptor 4 (TLR4) on proximal tubules, subsequently triggering interstitial inflammation. Vascular dysfunction, endothelial injury, immunological dysregulation, and abnormal cellular responses to injury collectively contribute to the development of AKI in sepsis [76].AKI resulting from major surgery can be attributed to fluid depletion, including blood loss and the extravasation of fluid into the third space [77]. Additionally, anesthetic agents may induce peripheral vasodilation and myocardial depression, thereby influencing renal perfusion. In the case of AKI associated with cardiac surgery, IR injury may occur due to extracorporeal circulation, leading to cell injury and death by increasing mitochondrial permeability [77,78]. Renal IR injury stands as the primary cause of AKI, contributing to tubular epithelial apoptosis, necrosis, and inflammation during the peri-operative period [79].

### 3.2. Intrinsic Causes

Nephrotoxic agents undergo filtration and concentration in the nephrons, potentially causing injury to renal TECs through direct cytotoxic effects. Furthermore, these toxins can lead to mesangial cell constriction by impairing intrarenal hemodynamics [72,80]. Some nephrotoxic agents may induce acute tubular injury. For instance, filtered polycationic aminoglycosides, which have a distinct affinity for the anionic phospholipid membranes, may interact with the megalin–cubilin receptors situated on the apical surfaces. This interaction facilitates the endocytosis of aminoglycosides, leading to their subsequent internalization into the cell and eventual translocation to lysosomes [81]. Lysosomal injury with myeloid body formation and mitochondrial injury results in tubular cell apoptosis and/or necrosis. Additionally, certain drugs, such as β-lactam antibiotics, proton pump inhibitors, and immunotherapies, can trigger acute interstitial nephritis [82]. These medications or their metabolites may initiate an immune reaction through different mechanisms. They can attach to the tubular basement membrane, acting as haptens or prohaptens, sparking an immune response against this antigen. Dendritic and tubular cells present the antigen to naive CD41 T-helper cells, leading to the formation of various T-helper cell subsets. These cells release cytokines, such as interleukins and interferons, which attract macrophages, eosinophils, CD8 T cells, and mast cells/basophils to the tubulointerstitium, ultimately causing acute interstitial nephritis [83].In individuals genetically predisposed to autoimmune activation, the renal consequences may involve glomerular inflammation and injury, such as rapidly progressive glomerulonephritis [84].

### 3.3. Post-Renal Causes

Extrarenal or intrarenal obstruction has the potential to elevate intratubular pressure, compromise renal blood flow, and trigger inflammatory processes, ultimately leading to AKI [85].

## 4. Molecular Mechanisms Involved in AKI to CKD Transition

### 4.1. Inflammation

While AKI is linked to the various mechanisms outlined earlier, it is primarily considered a complex clinical syndrome driven by inflammatory diseases with systemic effects [86]. Following an acute insult, stressed cells and injured tissues may release DAMPs, which interact with pattern recognition receptors (PRRs) such as TLRs to activate the innate immune pathway, resulting in the production of proinflammatory cytokines, chemokines, and reactive oxygen species (ROS), which eventually lead to further cell necrosis and tissue damage [87,88]. Intracellular molecules such as high mobility group box 1 (HMGB1), histones, heat kinin, and fibronectin released from necrotic renal tubular cells enter the extracellular space, exacerbating the inflammatory cascades [89]. Moreover, the sustained release of inflammation-associated fibrotic cytokines such as transforming growth factor-β (TGF-β) and interleukin-13 (IL-13) can trigger epithelial–mesenchymal transition (EMT), potentially leading to renal fibrosis and chronic renal insufficiency [90].

The most prominent signaling pathways involved in the expression of inflammation-associated genes include nuclear factor kappa-B (NF-κB), mitogen-activated protein kinase (MAPK), and STAT pathway components. NF-κB, a crucial nuclear transcription factor, plays a key role in regulating genes associated with the inflammatory response and influencing the release of inflammatory cytokines, chemokines, and adhesion factors. The NF-κB family comprises five related protein members: p50, p52, RelA (p65), RelB, and c-Rel. The inactivation of NF-κB is regulated by IκB kinase (IΚΚ) [91]. IκB undergoes phosphorylation and rapid degradation in response to stimulation by ROS and cytokines. This process results in the liberation of the free NF-κB dimer, which then undergoes phosphorylation and translocation to the nucleus. Subsequently, this translocation promotes the transcription of inflammation-related genes [92]. However, research has revealed that silent information regulator transcript 1 (SIRT1), also known as Sirtuin1, has the potential to mitigate kidney injury [93]. SIRT1 is a histone deacetylase, and overexpressing SIRT1 in renal TECs inhibits NF-κB activation. This inhibition occurs through the deacetylation of the Lys310 residue on the RelA/p65 subunit or by reducing the activity of the acetyltransferase P300/CBP [94]. Hence, the SIRT1 pathway presents a potential therapeutic target for mitigating inflammatory damage in AKI. Diosmin, a glycosylated polyphenolic flavonoid found in Citrus aurantium, attenuates renal fibrosis, mainly through an anti-inflammatory effect that is dependent on SIRT3-mediated nuclear expression of the NF-κB p65 [95]. Moreover, targeted therapies for the suppression of the inflammatory process in the context of kidney disease have gained attention. This includes the modulation of the NF-κB pathway and the specific suppression of NF-κB/NLRP3 activity by regulating the NF-κB signaling pathway using miRNAs [96].

### 4.2. Hypoxia

Renal tubular epithelium cells rely on ATP from the mitochondrial respiratory chain to maintain renal function through Na-K-ATPases, and this process is highly oxygen-dependent. Insufficient oxygen during AKI can lead to mitochondrial dysfunction, increased ROS production, and endothelial inflammation, which can further induce peritubular capillary rarefaction, worsening the tissue hypoxia and perpetuating this cycle [97]. Hypoxia was shown to significantly increase miR-493, leading to the suppression of stathmin-1 (STMN-1), a cell cycle regulator. This induction resulted in G2/M cell cycle arrest and the release of profibrotic cytokines in vitro [98]. Moreover, hypoxia-induced factors (HIFs) are also involved in the process of post-AKI renal fibrosis development. Normally, there are various subunits that comprise HIFs, including various α subunits (HIF1/2/3α) and a shared HIF1β. In normal physiology conditions, HIF-1α undergoes hydroxylation by prolyl hydroxylase domain (PHD) proteins, followed by binding to the Von Hippel–Lindau (VHL) E3 ubiquitin ligase and subsequent degradation in proteasomes [99]. Under hypoxia conditions, the inhibition of PHD proteins results in the translocation of HIF-1/2α into the cell nucleus, which binds with HIF-1β and further initiates gene transcription, including the expression of vascular endothelial growth factor (VEGF), erythropoietin (EPO), and glucose transporter 1 (GLUT1) [100]. Whether HIFs are protective factors for renal fibrosis is still under debate. Some studies have demonstrated that HIFs are participants in the regulation of pro-fibrotic genes and the promotion of renal fibrosis via TGF-β, NF-κB, and the phosphatidylinositol 3-kinase/protein kinase B (PI3K/Akt) pathway or via G2/M cell cycle arrest pathway through p53 upregulation [101,102]. On the contrary, some authors have found that under hypoxia conditions HIF-1α binds to FoxO3, which is thought to be a renoprotection factor after AKI, and subsequently inhibits the hydroxylation and degradation of FoxO3 [103]. SerpinA3K is another novel biomarker for the AKI to CKD transition in animal models, and knockout of serpinA3K resulted in higher FoxO3 expression with improved cellular responses to hypoxic injury, suggesting SerpinA3K’s involvement in the renal oxidant response, HIF1α pathway, and cell apoptosis [104,105]. Studies exploring the effectiveness of PHD inhibitors in renal injury highlighted the positive impact of HIFs on renal fibrosis [106]. Roxadustat (FG-4592), the pioneering HIF-PHI developed by FibroGen nearly a decade ago, functions by inhibiting PHD proteins, thereby modulating the dynamic between HIF synthesis and degradation [107]. This drug has gained approval for use in patients with renal anemia associated with CKD across numerous countries [108]. Additionally, studies have demonstrated that roxadustat impacts kidney injury by inhibiting inflammatory factors, attenuating mitochondrial injury, and reducing the levels of Bax and cleaved caspase-3, thereby decreasing apoptosis [109]. Moreover, it inhibits the renal fibrosis process by maintaining the redox balance and enhancing renal vascular regeneration. This effect is mediated through the HIF-1α/vascular endothelial growth factor A (VEGFA)/VEGF receptor 1 (VEGFR1) signaling pathway and by promoting the expression of the endogenous antioxidant superoxide dismutase 2 (SOD2) [110].

### 4.3. Signaling Pathways Involved in the Process of Renal Fibrosis

Maladaptive repair after renal tubule injury inducing progressive renal fibrosis and destruction of the normal architecture of the kidney is thought to be the key pathological mechanism contributing to CKD [111,112]. The transformation of various local stromal cells in the kidney to myofibroblasts plays important roles in the progressive kidney fibrosis, including effects on resident fibroblasts and pericytes/perivascular fibroblasts, EMT, endothelial to mesenchymal transition (EndMT), and macrophage (bone-marrow-derived) to myofibroblast transition (MMT) [113]. Myofibroblasts contribute to excessive extracellular matrix production and deposition in the renal parenchyma, which eventually lead to chronic kidney fibrosis and loss of renal function [16]. There are severe molecules involved in the complicated process of renal fibrosis, mainly the Wnt/β-catenin, TGF-β1/SMAD, and Hippo signaling pathways (Figure 1).

#### 4.3.1. Wnt/β-Catenin Signaling Pathway

The Wnt/β-catenin signaling pathway is heavily involved in the initiation and signal transmission of renal fibrosis [114,115]. Normally, Wnt/β-catenin activation is responsible for cellular repair and regeneration after an acute insult of renal tissue; however, the constant activation of this pathway can contribute to renal fibrosis and eventually induce CKD [116]. In normal physiology processes, a protein complex composed of five proteins inactivates the Wnt/β-catenin pathway through phosphorylation to prevent the overactivation of this pathway and renal fibrosis. However, in the pathologic state, the Wnt ligands bind to frizzled protein (FZD), LDL receptor-associated protein 5/6 (LRP5/6), and LRP, which are mainly derived from the cytoplasm of tubular cells, leading to reduced phosphorylation of β-catenin and changing it into its active form. After β-catenin translocates to the nucleus and binds to T cell factor/lymphoid enhancer factor (TCF/LEF) transcription factors, the downstream signal, including TGF-β1/SMAD signals, RAS, Snail1, Twist1, matrix metalloproteinase 7 (MMP-7), transient receptor potential canonical 6 (TRPC6), and plasminogen activator inhibitor-1 (PAI-1), leads to fibroblast activation [117,118]. The sustained expression of Wnt ligands eventually induces myofibroblast transformation and results in fibrosis [16] (Figure 2). Additionally, WNT/β-catenin signaling is involved in CKD-associated vascular calcification and mineral bone disease. The WNT/β-catenin pathway is tightly regulated, for example, by proteins of the DKK family. In particular, DKK3 is released by “stressed” TECs, which drives kidney fibrosis and is associated with a short-term risk of CKD progression and acute kidney injury [116].

#### 4.3.2. TGF-β1/SMAD Signaling Pathway

The TGF-β1/SMAD pathway is another important mechanism contributing to the transition to myofibroblasts and renal fibrosis [119]. TGF-β1 is produced in an inactive state, where it is bound to latency-associated peptide (LAP) and latent TGF-β1-binding protein (LTBP). Various triggers, such as ROS, have the capability to liberate TGF-β1 from LAP and LTBP, leading to the activation of TGF-β1. The active form of TGF-β1 binds to the type II TGF-β1 receptor (TβRII) and further binds to the type I TGF-β1 receptor (TβRI) and induces the phosphorylation of the SMAD2/SMAD3 complex, which activates SMAD4. SMAD4 induces the translocation of SMAD2/SMAD3 from the cytoplasm to the nucleus, and the complex eventually activates miRNA-21 and miRNA-192, eventually leading to extracellular matrix production and renal fibrosis [120]. In contrast, SMAD7, an inhibitory SMAD that is transcriptionally induced by SMAD3, regulates the function of SMAD3 and serves as a negative feedback mechanism of the TGF-β1/SMAD pathway [121,122]. In normal physiological conditions, there is an abundance of Smad7 to inhibit the TGF-β1/SMAD pathway through the degradation of TβRI via an ubiquitin–proteasome degradation mechanism [123]. In pathologic states, the overexpression of SMAD3 would induce Smad ubiquitination regulatory factor 1 (Smurf1), Smad ubiquitination regulatory factor2 (Smurf2), and arkadia, which degrade the SMAD7 protein [124,125]. This process further contributes to the profibrogenic process, induces the transition to myeloblasts, and induces renal fibrosis progression [126,127,128] (Figure 2).

#### 4.3.3. Hippo/Yes-Associated Protein (YAP)/Tafazzin (TAZ) Signaling

The Hippo pathway was first identified 20 years ago and is thought to be involved in cell growth, proliferation, and apoptosis. It plays a key role in regulating organ size, tissue regeneration, and tumor development [129]. Various physiological and pathological signals can induce the Hippo signaling pathway, including extracellular matrix (ECM) stiffness, cell polarity, and energy stress [130]. The upstream membrane receptor serves as a receptor for the extracellular growth inhibition signal. When the inhibitory signal binds to the receptor, the TAO kinase activates the Hippo pathway via phosphorylation of STE20-like serine/threonine kinase 1/2 (MST1/2), which forms a complex with the adaptor protein Salvador 1 (SAV1). Subsequently, this complex phosphorylates the large tumor suppressor (LATS1/2) and the LATS1/2-interacting protein MOB kinase activator 1 (MOB1), and then the phosphorylated LATS1/2–MOB1 complex phosphorylates YAP and TAZ, which promotes the cytoplasmic polyubiquitination and consequent degradation of YAP/TAZ by the proteasome [131]. On the contrary, inactivation of the Hippo pathway results in the dephosphorylation of upstream kinases, causing the active YAP/TAZ to migrate into the nucleus. There, they interact with various transcription factors, including TEA domain DNA-binding family members (TEAD1–4), to regulate cell proliferation [132]. YAP/TAZ would also bind to other transcription factors, including TCF/LEF transcription factors, SMAD1, SMAD2/3, and p37 [133] (Figure 3).

Studies have demonstrated that the Hippo pathway is associated with AKI [134]. YAP protein levels rise in both the cytoplasm and nucleus of renal TECs during the AKI repair stage, and they are also positively correlated with the changes in YAP and TEAD expression [135]. Moreover, the Hippo pathway is linked to EMT in renal TECs, a pivotal process in the transition from AKI to CKD. EMT occurs through the activation of the TGF-β/Smad pathway and loss of tubular epithelial cell polarity. This polarity is primarily maintained by the Crumbs (CRB)/PALS1 complex, which also regulates the Hippo signaling pathway [136,137]. The research has indicated that traditional Chinese medicines may influence the Hippo pathway and regulate renal fibrosis, although clinical evidence for their use in kidney diseases remains insufficient [133]. Triptolide, a diterpenoid trioxide compound, has been found to inhibit EMT in renal tubular cells and reduce renal fibrosis through modulation of the Hippo signaling pathway [138]. Additionally, quercetin, a biological flavonoid, activates the Hippo signaling pathway, potentially slowing kidney disease progression and aiding in the prevention of renal fibrosis [132]. Another compound, Verteporfin (VP), a photosensitizer used in treating age-related macular degeneration, can bind to YAP, disrupting its interaction with TEAD, which might improve renal tubulointerstitial inflammation and fibrosis [139,140].

### 4.4. Innate and Adaptive Immunity

Except for myofibroblast transition, the immune system also plays a crucial role in all aspects of the pathophysiology of kidney injury and repair. In the acute phase of injury, components of the innate immune system, such as macrophages and neutrophils, are recruited to the injured site. They release pro-inflammatory cytokines such as interleukin-6 (IL-6) and tumor necrosis factor-alpha (TNF-α), thereby amplifying the inflammatory response. At the later stage of AKI, the immune system is also involved in the repair process. For example, macrophages (which present as the M1 phenotype during the acute insult, exacerbating the inflammation process by releasing interleukin-1 (IL-1), IL-6, and TNF-α) switches to the M2 phenotype to mediate the inflammation cascade and propagate the repair process [141,142]. However, M2-type macrophages are implicated in renal fibrosis via MMT or numerous growth factors [127]. TECs release Wnt ligands, prompting a transition in the macrophage phenotype from M1 to M2, thereby promoting fibrosis [143]. Macrophages also secrete Wnt proteins, TGF-β1, and TIMPs, exerting a pivotal influence on kidney fibrosis through their involvement in the synthesis and deposition of extracellular matrix [144].

Increasing complement levels in kidney cells is also implicated in the pathogenesis of kidney fibrosis. For example, elevated C1q levels in PDGFRβ-positive pericytes lead to heightened inflammation and renal scarring. This is attributed to an increased production of cytokines such as IL-6, monocyte chemoattractant protein-1 (MCP-1; CCL2), and macrophage inflammatory protein-1 alpha (MIP1-α; CCL3) [145]. Both mRNA and protein expression of C1r and C1s are also increased during fibrosis, and this contributes to the increasing level of C3 fragments, which eventually propagate the downstream reaction of the complement system and lead to the activation of myofibroblast transition and renal fibrosis [146]. Animal studies have demonstrated that C5 knockout and C5R antagonist usage can reduce tissue fibrosis [147]. Additional studies have demonstrated that the inhibition of C1r serine protease or C3a/C3aR can effectively alleviate interstitial fibrosis in diverse murine models [148,149].

The adaptive immune system is also engaged in the AKI process. During the initial insult, renal dendritic cells present antigens to T cells. Activated T cells subsequently release pro-inflammatory cytokines, including interferon-gamma (IFN-γ), contributing to the overall inflammatory cascade [150]. Much like the observed shift in phenotype seen in macrophages, regulatory T cells, which are powerful mediators of the immune system, become discernible in the kidney. These regulatory T cells suppress the activation of many different immune cells through both contact-dependent mechanisms and the release of soluble mediators, such as interleukin-10 (IL-10), which induce the phosphorylation of STAT1, 3, and 5, thereby promoting the repair and anti-fibrotic process following injury [151,152,153].

### 4.5. Mitochondrial Dysfunction

AKI, especially ischemic injury, can lead to mitochondrial dysfunction because hypoxia disrupts the electron transport chain in mitochondria and produces excessive free-radical-containing substances. These harmful molecules further damage tubular cells [154]. In recent studies, mitochondrial dysfunction has also been increasingly recognized as a crucial contributor in the process of AKI to CKD transition [16,155]. Basically, mitochondrial homeostasis is maintained through three processes: mitochondrial dynamics, mitophagy, and mitochondrial biogenesis [156]. Mitochondrial dynamics includes two different processes: fission (regulated by DRP1) and fusion (regulated by MFN1, MFN2, and OPA1). Mitophagy is the process in which damaged mitochondria are selectively degraded by autophagy, which is regulated by the PINK1–PARK2 pathway, the BNIP3 and NIX pathways, and the FUNDC1 pathway. Mitochondrial biogenesis, which is mainly regulated by peroxisome proliferator-activated receptor γ coactivator -1α (PGC-1α), addresses the heightened cellular energy demands and replenishes the mitochondrial content in newly formed cells during the process of cell proliferation [154,157]. Animal studies have demonstrated that adjusting these genes involved in the process of mitochondria homeostasis influences the process of kidney inflammation, kidney fibrosis, and tubular epithelial cell apoptosis [79,158,159,160]. For instance, elevated PGC-1α levels in tubular cells boost the mitochondrial mass, providing kidney protection post-ischemic and inflammatory injury without cell death [161]. Conversely, global PGC-1α deficiency leads to severe renal dysfunction in septic AKI [160].

### 4.6. G2/M Arrest Pathway and Cellular Senescence

The initiation of DNA damage response (DDR) signaling is essential for the reparative mechanisms in renal proximal epithelial cells following AKI. When the repair is incomplete, the injured cell is arrested at the G2/M phase to stabilize genetic factors [162,163]. These cells subsequently exhibit a marked upregulation in the messenger RNA expression of fibrogenic growth factors, such as TGF-β1 and connective tissue growth factor (CTGF), which eventually leads to interstitial fibrosis [112,164]. The targets of rapamycin (TOR)-autophagy special coupling compartments (TASCCs), novel complexes that are present in G2/M-arrested tubular cells, are associated with cellular senescence [165]. Cyclin G1 (CG1) plays a pivotal role in the formation of TASCCs, and studies have indicated that the deletion of CG1 or the specific deletion of raptor (a key component of the TASCC complex) in proximal tubules markedly mitigated renal fibrosis. Epithelial Toll-like and interleukin 1 receptors (TLR/IL-1Rs) are other factors that mediate cellular senescence and the G2/M arrest pathway [17]. After AKI, the activation of TLRs by IL-1 and DAMPs can trigger an excessive inflammatory response and facilitate interstitial fibrosis; in contrast, the deletion of Myd88, an TLR/IL-1 downstream protein and an NF-κB upstream protein, ameliorates renal fibrosis after kidney damage [17,166]. Hence, targeting tubules experiencing G2/M cell-cycle arrest represents a promising therapeutic approach, with several interventions having been documented. For example, pifithrin-α, an inhibitor of the crucial cell cycle regulator p53, is a promising drug candidate that could be used after acute kidney injury for reducing post-AKI renal fibrosis [167]. The PTBA analog methyl-4-(phenylthio)butanoate (M4PTB), a histone deacetylase inhibitor, has also been demonstrated to decrease G2/M arrest in injured tubular cells, thereby reducing interstitial fibrosis [168]. Other studies have revealed the effectiveness of specific inhibitors targeting cyclin-dependent kinases 4/6, pivotal mediators of cell-cycle progression from the G1 to the S phase, in optimizing the cell-cycle progression of injured tubules [169,170].

### 4.7. Renin—Angiotensin—Aldosterone System

The RAAS is initiated by the release of renin from juxtaglomerular cells in the kidneys. It converts the liver-formed angiotensinogen into angiotensin I, which subsequently transforms into angiotensin II through the action of angiotensin-converting enzymes. Prolonged and overactivation of RAAS after AKI can lead to CKD progression through various mechanisms [171]. Angiotensin II binding to angiotensin II receptor type 1 (AT1R) activates the RhoGEF/RhoA/ROCK cascade and induces overexpression of NF-κB, PAI-1, MAPK/extracellular signal-regulated kinases (ERK) 1/2, and NADPH oxidases. NADPH oxidases, ROS generating enzymes, increase the oxidative stress inside the cell and contribute to the overexpression of TGF-β/SMAD and MAPK/ERK1/2 [172,173]. All these molecules can contribute to organ remodeling tissue fibrosis. Moreover, AT1R activation mediates the balance between the intracellular levels of nitric oxide (NO) and calcium through the PI3K/Akt pathway. The suppression of endothelial nitric oxide synthase (eNOS) and increasing cytosol calcium levels through inositol 1,4,5-trisphosphate (IP3) elevate the resistance of efferent arterioles and disrupt the autoregulation of afferent arterioles, which eventually leads to glomerular hyperfiltration and sclerosis [172,174,175] (Figure 3). Aldosterone is a mineralocorticoid hormone that is thought to be involved in the process of renal fibrosis independent from its actions to increase blood pressure by mediating salt retention. Aldosterone exerts its influence on the kidney via inducing the production of ROS, upregulating the expression of EGFR and AT1R, and activating NF-κB and activator protein-1 (AP-1). These molecular events further contribute to cell proliferation, apoptosis, and the phenotypic transformation of epithelial cells, which further triggers the expression of TGF-β, CTGF, and PAI-1, ultimately culminating in the development of renal fibrosis [176,177].

Numerous clinical trials support the renoprotective effects of RAS inhibitors such as ACE inhibitors and AT1a receptor blockers in diabetic or proteinuric non-diabetic CKD patients [178,179]. A meta-analysis involving 70,801 patients revealed that exposure to ACEi/ARB after AKI is associated with lower risks of recurrent AKI and progression to incident CKD [180]. However, there are only a few observational studies that explored the impact of RAS inhibitors on the AKI to CKD transition. Moreover, there has not been a large-scale randomized controlled trial to assess the impact of RAS blockade on AKI and the subsequent development of CKD. The role of RAS activity in the acute phase and severity of AKI is still uncertain.

## 5. Conclusions

The classification of kidney disease into AKI, AKD, and CKD highlights the risks of progression and the causal relationship between AKI/AKD and CKD. It addresses the need for better biomarkers for early AKI detection, beyond traditional serum creatinine, and the complex molecular pathways involved in the AKI to CKD transition. The biomarkers for the early detection of AKI are categorized into stress, damage, and functional markers. AKI emerges after diverse insults, leading to complex pathophysiological events such as cellular hypoxia, inflammation, and nephrotoxicity, with subsequent renal damage. The inflammatory response plays a central role in AKI, where cellular stress leads to the release of DAMPs and activation of innate immunity, causing further injury and fibrosis. Molecular pathways including NF-κB, MAPK, and STAT are pivotal in the inflammation and fibrosis associated with AKI. Maladaptive repair mechanisms after AKI, involving the transition of various renal cells to myofibroblasts, contribute significantly to the progression of CKD. The Wnt/β-catenin, TGF-β/SMAD, and Hippo signaling pathways are critical in this transition, promoting fibrosis and affecting renal function. Mitochondrial dysfunction, cellular senescence, hypoxia, and the RAAS are also key contributors to AKI progression and CKD transition, with the RAAS in particular playing a role in renal remodeling and fibrosis. Despite the recognized importance of the RAAS, more research is needed to fully understand its role in AKI and the subsequent CKD development.

## Figures and Tables

**Figure 1 ijms-25-01755-f001:**
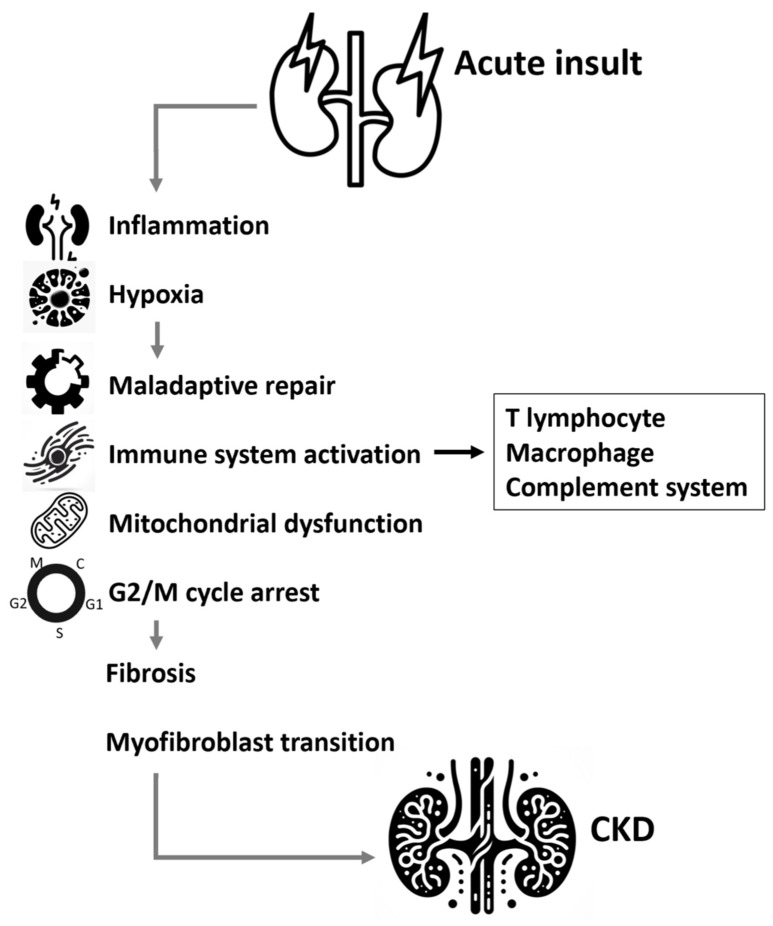
Mechanism involved in the transition from acute kidney injury to chronic kidney disease. Abbreviation: CKD, chronic kidney disease.

**Figure 2 ijms-25-01755-f002:**
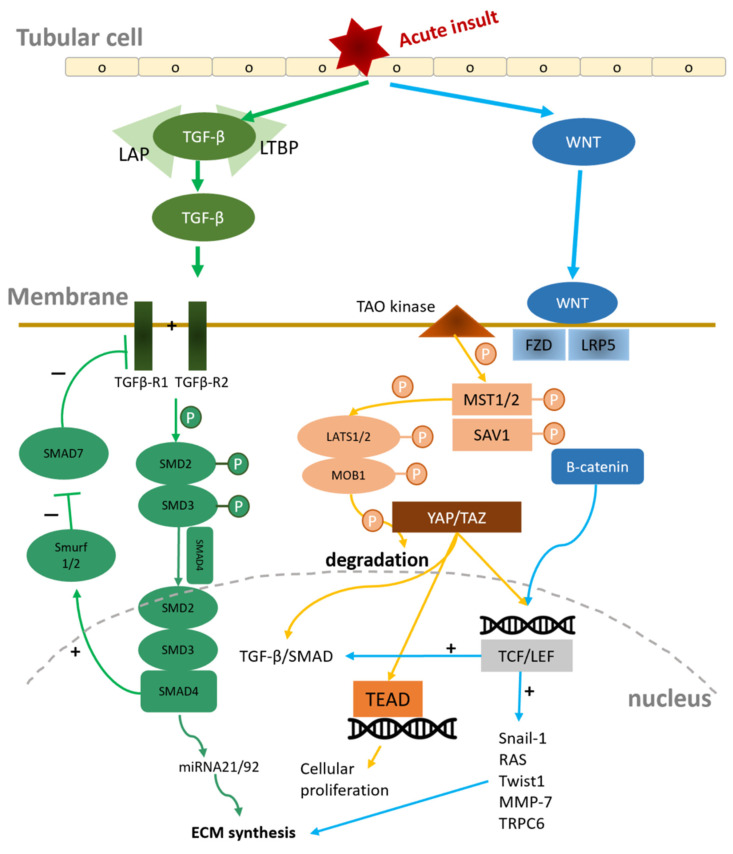
(1) Under normal conditions, in the absence of Wnt ligand interactions, LRP contributes to the phosphorylation of β-catenin, resulting in its retention in the cytoplasm. However, during pathological processes, Wnt ligands bind to the FZD/LRP complex, leading to the prevention of β-catenin phosphorylation. This allows β-catenin to migrate into the nucleus and initiate downstream pathways, thereby promoting renal fibrosis. (2) After an acute insult, TGF-β1 becomes activated and binds to its receptor, which in turn phosphorylates SMAD2/3. The phosphorylated SMAD2/3, along with SMAD4, then translocates into the nucleus to activate the expression of miRNA-21 and miRNA-192. This activation ultimately leads to renal fibrosis. Concurrently, Smurf1/2 is activated by the SMAD2/3/4 complex, which diminishes the inhibitory capability of SMAD7. (3) The activation of the Hippo pathway leads to the phosphorylation of MST1/2, SAV1, LAST1/2, and MOB1, which contributes to the degradation of YAP/TAZ. Conversely, the inactivation of the Hippo pathway results in the activation of YAP/TAZ, allowing this complex to migrate into the nucleus. This migration initiates cellular proliferation and contributes to the development of renal fibrosis. Abbreviations: FZD, frizzled protein; LATS1/2, large tumor suppressor; LAP, latency-associated peptide; LRP, LDL receptor-associated protein; LTBP, latent TGF-β1 binding protein; MMP-7, matrix metalloproteinase 7; MOB1, MOB kinase activator 1; MST1/2, STE20-like serine/threonine kinase 1/2; P, phosphorylated; SAV1, Salvador 1; Smurf, Smad ubiquitination regulatory factor; TAZ, tafazzin; TCF/LEF, T cell factor/lymphoid enhancer factor transcription factor; TEAD1–4, TEA domain DNA-binding family members; TGF-β1, transforming growth factor-β; TRPC6, transient receptor potential canonical 6; YAP, yes-associated protein.

**Figure 3 ijms-25-01755-f003:**
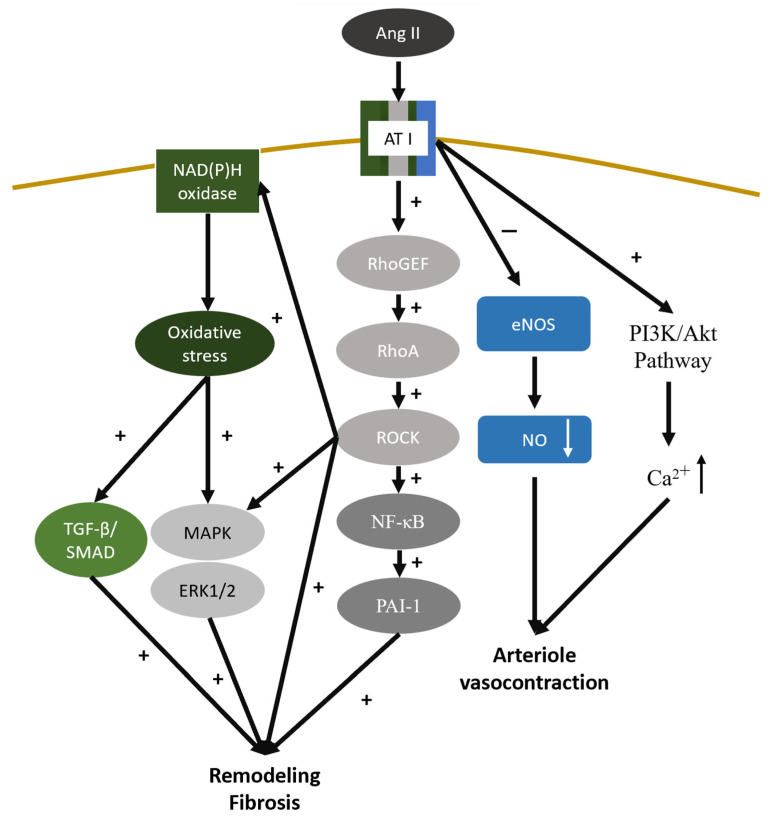
Binding of angiotensin II to the AT1 receptor activates the RhoGEF/RhoA/ROCK cascade, leading to overexpression of NF-κB, PAI-1, MAPK/ERK 1/2, and NADPH oxidases. NADPH oxidases, as ROS-generating enzymes, increase oxidative stress, thereby upregulating the TGF-β/SMAD and MAPK/ERK pathways. This cascade results in organ remodeling and tissue fibrosis. Additionally, AT1R activation alters intracellular NO and calcium levels via the PI3K/Akt pathway and disrupts the autoregulation of the kidney through arteriole vasocontraction. A downward arrow indicates a decrease in the substance level, while an upward arrow signifies an increase. Abbreviations: AT1R, angiotensin II receptor type 1; eNOS, endothelial nitric oxide synthase; ERK, extracellular signal-regulated kinase; MAPK, mitogen-activated protein kinase; NF-κB, nuclear factor kappa-B; NO, nitric oxide; PAI-1, plasminogen activator inhibitor-1; PI3K/Akt, phosphatidylinositol 3-kinase/protein kinase B; TGF-β, transforming growth factor-β.

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
