# Peer review of "From Acute to Chronic: Unraveling the Pathophysiological Mechanisms of the Progression from Acute Kidney Injury to Acute Kidney Disease to Chronic Kidney Disease"

_ijms, 2024, doi:10.3390/ijms25031755_

Round 1
Reviewer 1 Report
Comments and Suggestions for Authors
The authors provide a comprehensive overview of the continuum from AKI to CKD, highlighting biomarkers for early AKI detection as well as the complex molecular mechanisms underlying this transition. It outlines various categories of AKI biomarkers, including markers of renal stress, damage, and function, which can facilitate early diagnosis, prognostication, and clinical management. The pathophysiology of AKI is also discussed, arising from diverse insults like decreased renal perfusion, sepsis, toxins or surgery, eliciting tissue hypoxia, inflammation, and direct tubular injury. At a molecular level, inflammation is pivotal in AKI, mediated by pathways like NF-κB, MAPK, and STAT which trigger cytokine release and oxidative stress. Maladaptive repair processes then promote renal fibrosis and dysfunction, with key roles for signaling pathways like Wnt/β-catenin, TGF-β/SMAD, and Hippo/YAP/TAZ. Other processes like mitochondrial dysfunction, cellular senescence, G2/M cell cycle arrest, and Renin-Angiotensin-Aldosterone System activation also drive these deleterious changes underlying the AKI to CKD transition. This comprehensive review analyzes biomarkers, pathophysiology and molecular mechanisms spanning the spectrum from AKI to eventual CKD. It highlights the central role of inflammation and microvascular loss in AKI, leading to aberrant tissue repair responses promoting progressive nephron loss. Understanding these intricate molecular pathways offers insights into novel biomarkers and rational therapeutic targets to mitigate AKI and impede advancement to CKD.
Comments
- The article discusses AKI in broad terms without clearly defining specific subtypes like prerenal AKI from hypovolemia, intrinsic AKI from sepsis/nephrotoxins/ischemia or postrenal AKI from urinary tract obstruction. Providing clear classifications and descriptions for each distinct form of AKI based on underlying pathophysiological mechanism would allow for more targeted and granular analysis when reviewing biomarkers or molecular pathways involved.
- There is a missed opportunity to discuss detection of subclinical AKI based on novel biomarkers reflecting kidney stress and damage prior to loss of function. Incorporating evidence and markers able to identify AKI early in evolution or spot mild forms that evade creatinine-based criteria would significantly strengthen the manuscript. Examples include using cell cycle arrest biomarkers like TIMP-2 and IGFBP7 to reveal covert AKI not flagged by standard tests.
- While current AKI biomarker classification schemes are discussed, incorporating the more recent AKI "Snapshot" which designates markers as "Rise", "Damage", "Response" or "Repair" would ensure contemporary relevance given research progress in the interim.
- Emerging novel AKI biomarkers like angiotensinogen, methylguanidine or meprin-α with growing evidence supporting utility could be analyzed to provide forward-looking insights alongside traditional markers. Their exclusion overlooks important progress in AKI diagnostics.
- Quantitative metrics assessing clinical validity like sensitivity, specificity, AUC-ROC characteristics for referenced biomarkers in AKI prediction or prognosis are absent. Incorporating available performance data would better showcase their diagnostic and predictive value.
- Pharmacologic interventions for impacted pathways are mentioned briefly but not expanded upon. Further discussing completed or active clinical trials targeting elements like HIF, NF-kB or Hippo would significantly bolster the treatment relevance of delineated biology underlying AKI-CKD transitions.
Reviewer 2 Report
Comments and Suggestions for Authors
The authors have made an interesting review about the molecular mechanisms underlying the transition process from AKI to CKD. The review is quite complete and has a coherent structure.
Regarding biomarkers I think it would be interesting to have a specific comparative section between usual clinical practices and the biomarkers they describe, in order to identify the best potential candidates. In addition, it would also be interesting to include what methodology would be necessary to use, and whether the costs would be commensurate with the clinical context.
Also consider including future perspectives, i.e., a critical analysis of the evidence described and what should be prioritized in future research.
Kind regards
